# Key Factors in Early Diagnosis of Myopia Progression within Ocular Biometric Parameters by Scheimpflug Technology

**DOI:** 10.3390/life13020447

**Published:** 2023-02-04

**Authors:** Alfredo López-Muñoz, Beatriz Gargallo-Martínez, María Carmen Sánchez-González, Raúl Capote-Puente, Concepción De-Hita-Cantalejo, Marta Romero-Luna, Juan-José Conejero-Domínguez, José-María Sánchez-González

**Affiliations:** 1Department of Physics of Condensed Matter, Optics Area, Vision Sciences Research Group (CIVIUS), Pharmacy School, University of Seville, 41009 Seville, Spain; 2Research & Development Department (Miranza Virgen de Luján®), Ophthalmology Center, 41011 Seville, Spain

**Keywords:** axial length, ocular biometry, early diagnosis, myopia progression, Scheimpflug technology

## Abstract

The aim of this study was to evaluate the relationship between myopia and ocular biometric variables using the Pentacam AXL^®^ single rotation Scheimpflug camera. This prospective, cross-sectional, single-center study was performed in fifty Caucasian patients aged between 18 and 30 years (24.84 ± 3.04 years). The measured variables included maximum and minimum keratometry (K1 and K2, respectively), anterior chamber depth (ACD), corneal horizontal diameter or white to white (WTW), central corneal thickness (CCT), corneal asphericity (Q), and axial length (AXL). The tomographic and biometric measurements were considered optimal when the quality factor was greater than 95% according to the manufacturer’s software instructions. The AXL presented a significant correlation with the spherical equivalent without cycloplegia (SE without CP), age at onset of myopia (r = −0.365, *p* = 0.012), mean keratometry (Km) (r = −0.339, *p* = 0.016), ACD (r = 0.304, *p* = 0.032), and WTW (r = 0.406, *p* = 0.005). The eyes with AXL higher than 25 mm had earlier onset; higher SE without CP, AXL, and Q; and a flatter Km. AXL is the biometric variable with the greatest influence on the final refractive state in the adult myopic eye. Ophthalmologists and optometric management must consider these biometric differences in order to identify the most appropriate correction techniques in each case. The use of the Pentacam AXL in ocular biometric measurement is effective, reproducible, and non-invasive.

## 1. Introduction

Myopia is the most common clinically significant refractive defect and is responsible for 5–10% of cases of legal blindness in developed countries [1]. Myopia, also known as nearsightedness, is a prevalent public health concern that can lead to visual impairment and increase the risk of other serious eye conditions. It is widely recognized as a significant issue [2]. Epidemiological studies conducted in recent years have detected a rising prevalence of myopia globally. Some projections estimate myopia will affect 34% of the world’s population in 2020, with an asymmetric distribution between different geographical regions and ethnic groups. In Western Europe, according to some publications, myopia affects approximately 30–35% of people in 2020 [1]. Myopia is often first diagnosed in childhood and can progress throughout adolescence. It is more common in people of East Asian descent and tends to run in families. In some countries in these regions, over 80% of the population has myopia. The prevalence of myopia is also increasing in other parts of the world, including the United States and Europe [3].

There are several risk factors for the development of myopia, including genetics, increased near work (e.g., reading, using a computer), less time spent outdoors, and higher levels of education. Myopia is associated with a number of health complications, including an increased risk of retinal detachment, glaucoma, and cataracts. It can also lead to visual impairment and reduced quality of life [4]. In Spain, the occurrence of myopia appears to be on the rise. It is believed that lifestyle factors may be contributing to the increased likelihood of developing myopia [5]. Several studies suggest the importance of understanding the mechanisms responsible for the progression of myopia, and effective treatments that may slow or prevent the onset of myopia in children are being researched [6].

Axial length, or the length of the eye, is considered to be a significant factor in the development of myopia. When the length of the vitreous chamber (a fluid-filled space in the eye) exceeds the focal length of the eye’s optical components, it can increase the refractive power of the eye and contribute to the development of refractive errors, such as myopia [7]. Genetic factors play a significant role in the development and progression of myopia, which are also directly related to environmental factors of all activities carried out requiring near vision [8]. There are four structures in the eye that contribute to its refractive status: the cornea, aqueous humor, lens, and vitreous humor. When these structures do not work together properly, it can lead to refractive errors such as myopia. Factors such as corneal curvature, anterior chamber depth, lens thickness, vitreous chamber depth, and axial length are often studied in relation to eye diseases, as they can impact the coordination of the ocular components and contribute to the development of refractive disorders [4].

There have been recent advances in the development of devices for the measurement of ocular biometrics, which are essential for studying and controlling myopia [9,10]. These devices allow for the precise and non-invasive measurement of various ocular parameters, such as axial length, corneal curvature, and lens thickness. There is ongoing development of devices to more accurately measure ocular biometrics such as corneal curvature and lens thickness. Currently, several non-invasive devices are used in clinical settings to measure these parameters, including partial coherence interferometry, optical low coherence reflectometry, and swept source biometry. These devices allow for the analysis of relationships between different ocular components and facilitate the use of different measurement systems [11].

When talking about a clinical diagnostic test, parameters such as sensitivity, specificity, and positive and negative predictive values are described. These reflect the characteristics of a diagnostic test and are used to decide when they should be used (sensitivity and specificity of a test) or what meaning a test result has in a particular patient.

Sensitivity is the probability of correctly classifying patients or, what is the same, the proportion of true positives, while the specificity is the probability of correctly classifying the healthy ones or, what is the same, the proportion of true negatives. Accordingly, the sensitivity and specificity represent the validity of a diagnostic test and the positive predictive value and negative predictive value represent the safety of a diagnostic test [12,13].

The Pentacam AXL^®^ device, made by Oculus in Germany, combines the Scheimpflug principle with partial coherence interferometry to measure ocular biometrics, including anterior segment tomography, axial length, and intraocular lens calculations and it is an indispensable tool, with high values of sensitivity and specificity in diagnosis. It uses a blue LED light source with a wavelength of 475 nm and a 1.45-megapixel camera to capture images of the cornea and record 138,000 datapoints in 2 s. Keratometry is calculated using a reference surface. This device builds on the proven measurement method of the Pentacam HR [14]. Other new devices for the measurement of ocular biometrics include swept source biometry and optical low coherence reflectometry. These devices use different technologies to measure ocular parameters, but they have also been found to be accurate and reliable. The use of these new devices for the measurement of ocular biometrics has allowed for a greater understanding of the factors that contribute to the development and progression of myopia. This knowledge can be used to develop effective interventions and treatments for myopia control [11].

The purpose of this study was to examine the connection between myopia and ocular biometric measurements taken using the Pentacam AXL^®^ device, a single rotation Scheimpflug camera with version 6.08r19 software produced by Oculus Optikgeräte in Germany.

## 2. Materials and Methods

### 2.1. Design

This research was conducted as a prospective, cross-sectional, single-center study at the University of Seville’s School of Pharmacy in Spain from February to April 2019. It followed the guidelines of the Declaration of Helsinki and was approved by the Ethical Committee Board of Andalusia. All study participants provided informed consent after being informed about the nature of the study.

### 2.2. Subjects

Fifty Caucasian patients aged 18 to 30 years (mean age: 24.84 ± 3.04 years) who were students at the University of Seville were recruited for the study [15]. To eliminate bias, only one eye of each participant was randomly chosen for inclusion. The inclusion criteria were as follows: being between 18 and 30 years old, having a stable refractive error (no more than 0.50 D change in spherical and cylindrical refraction in the past year), having a simple or compound myopic refractive error with or without astigmatism, having a corrected visual acuity of at least 20/25 in both eyes, and not wearing contact lenses for at least 2 weeks (4 weeks for hard lenses). The exclusion criteria were as follows: having any eye disease (e.g., glaucoma, cataracts), progressive corneal disease (e.g., keratoconus, pellucid marginal degeneration), corneal dystrophy or degeneration, cataracts or sclerosis of the lens, a current or previous history of uveitis, dry eye syndrome, persistent epithelial defects, central corneal leucoma, being on antiglaucoma or hypotonic therapy, signs of retinal vascular pathology, being pregnant or lactating, and having disorders of the eye muscles (e.g., strabismus, nystagmus) or any other disorder that affects ocular fixation.

### 2.3. Procedure

The participants underwent cycloplegic autorefraction and traditional refraction using retinoscopy and then subjective refraction with fogging was performed. Fogging refers to using plus powers to bring the optical point of focus in front of the retina to ensure that accommodation is adequately relaxed. The principle of fogging involves using spherical powers to create artificial myopia, thereby moving the entire area of focus in the eye in front of the retina to create a situation where an attempt at accommodating will blur the vision, which further causes the patient to relax accommodation. Fogging is effective irrespective of the inherent refractive state of the eye and the efficacy of fogging in refraction has been demonstrated [16].

Corneal tomography, analysis of the anterior segment, and measurement of axial length using the Pentacam AXL^®^ device were then carried out. The variables studied were maximum and minimum keratometry, anterior chamber depth (ACD), corneal horizontal diameter (WTW), central corneal thickness, corneal asphericity (Q), and axial length (AXL). The mean keratometry (Km) was calculated as the average of K1 and K2 within the 3-mm central optical zone for statistical analysis. A flowchart of the design of the study is presented in Figure 1.

Each measurement was taken three times in succession by the same experienced examiner under standardized conditions to minimize error. The tomographic and biometric measurements were considered optimal when the quality factor was greater than 95% according to the manufacturer’s software instructions.

### 2.4. Data Analysis

Statistical analysis was performed using SPSS software, version 23.0 (SPSS Inc., IBM, Chicago, IL, USA). Data distribution normality was studied by the Shapiro–Wilk test. Parametric variables (age, onset of myopia, AXL, Q, pachymetry, and ACD) and non-parametric variables (SE, Km, and WTW) were correlated using the Pearson coefficient correlation test and rho Spearman test, respectively. The sample was divided into two groups based on the dependent variables. The cutoff values (SE without CP, −4.50 D; AXL, 25 mm; age at myopia onset, 11 years) were arbitrarily set based on the median value. The mean difference in variables between these groups was analyzed using the independent samples Student’s *t*-test (parametric), Mann–Whitney U test (non-parametric), or Chi-Square test (categorical variables). Finally, a multiple stepwise linear regression was performed. *p* ≤ 0.05 was considered statistically significant.

## 3. Results

Fifty eyes from 50 patients (mean age, 24.84 ± 3.04 years, range 18–30 years) were included. There were 21 men (42%) and 29 women (58%) and no differences were found between sex groups for any variable analyzed. The descriptive data for the whole sample are presented in Table 1. The SE without CP showed a strong significant correlation with AXL (Figure 2) and age at myopia onset (Figure 3). Eyes with myopic SE without CP higher than −4.5 D had significantly longer AXL and younger age at myopia onset (Table 2).

No other ocular variables showed significant differences between SE without CP groups. The AXL was significantly correlated with the SE without CP (Figure 1), age at myopia onset (r = −0.365, *p* = 0.012), Km (r = −0.339, *p* = 0.016), ACD (r = 0.304, *p* = 0.032), and WTW (r = 0.406, *p* = 0.005). The eyes with AXL greater than 25 mm showed an earlier onset; higher values of SE without CP, AXL, and Q; and a flatter Km (Table 2). Age at myopia onset was significantly correlated with SE without CP (Figure 2), AXL (r = −0.365, *p* = 0.012), and Q (r = 0.371, *p* = 0.010). Patients with onset of myopia earlier than 11 years had higher SE without CP, AXL, and Q values (Table 2).

The multiple linear regression results showed that the variables most closely related to SE were AXL (*p* < 0.001, partial regression coefficient B = −2.441; standardized coefficient beta = −1.066), Km (*p* < 0.001, partial regression coefficient B = −1.021, standardized coefficient beta = −0.472), and ACD (*p* < 0.001, partial regression coefficient B = 3.616, standardized coefficient beta = 0.353); adjusted R^2^ = 0.917. A second multiple linear regression for each SE without CP group was performed, and different results were found. In the lower SE without CP group (SE > −4.5 D), a limited model (adjusted R^2^ = 0.474) showed the most closely related variable to SE without CP was the age at myopia onset (*p* = 0.011, standardized coefficient beta = 0.502), followed by the Km (*p* = 0.015, standardized coefficient beta = −0.478). In these cases, the AXL score was not significantly related (*p* = 0.232).

However, in the higher myopic SE without CP group (SE ≤ −4.5 D), the best model (adjusted R^2^ = 0.877) showed that the most closely related variables to SE without CP were AXL (*p* < 0.001, standardized coefficient beta = −1.102), Km (*p* < 0.001, standardized coefficient beta = −0.530), and ACD (*p* < 0.001, standardized coefficient beta = 0.455).

## 4. Discussion

The purpose of this study was to evaluate the relationship between myopia and ocular biometric variables using the Pentacam AXL single rotation Scheimpflug camera. This non-invasive measurement instrument was used to assess a range of variables, including maximum and minimum keratometry, anterior chamber depth, corneal horizontal diameter, central corneal thickness, corneal asphericity, and axial length. The results of the study showed that axial length (AXL) was significantly correlated with a number of other variables, including the spherical equivalent, age at onset of myopia, mean keratometry, anterior chamber depth, and corneal horizontal diameter. The eyes with AXL values higher than 25 mm were found to have earlier onset of myopia, higher spherical equivalent, AXL, and corneal asphericity values, and flatter mean keratometry values. These findings suggest that AXL is a key biometric variable that should be considered in the management of myopia, as it has a significant influence on the final refractive state of the adult myopic eye.

In recent years, there has been great interest in precisely knowing the dimensions of the human eye. Several studies of ocular biometrics have been conducted, but most have been performed in the elderly [10,17] or in subjects with a wide range of ages [18,19]. To minimize the impact of aging on ocular biometric measurements, it is necessary to carefully design studies that take this factor into account [20]. There have been numerous studies that have compared the values of variables in the anterior segment of myopic and emmetropic eyes. These studies have consistently found significant correlations between refractive errors and anterior chamber depth [17], corneal diameter [18,21], Km [22], and PQT [23]. The strength of correlations between refractive errors and anterior chamber depth may vary depending on the race and age of the study subjects. It is also known that the prevalence of myopia can differ based on age, ethnicity, and educational level, with higher rates being reported in university students in Asian countries [24]. To the best of our knowledge, there are no published studies on the correlations between the degree of myopia and the biometrics of the anterior segment in young Spanish university students.

Most studies that analyze university populations focus on the analysis of the prevalence of refractive errors by measuring only refraction [25], or only measuring AXL [26], while others use two devices for different measurements in the analysis of different ametropias [20]. One of the advantages of our study is the use of a single, non-invasive device, with which the different variables are obtained in a single measurement, such as the Pentacam AXL^®^, whose values of AXL, Km, and ACD have been shown to be highly repeatable [27].

There are various tools available for the measurement of axial length, including ultrasound [28], partial coherence laser interferometry [29], and optical coherence tomography [30]. Each of these tools has its own unique advantages and disadvantages. Ultrasound is a non-invasive method of measuring axial length, but its accuracy can be affected by factors such as the thickness of the cornea, presence of cataracts, and the presence of media opacities [28]. On the other hand, partial coherence laser interferometry provides highly accurate measurements, but requires a dilated pupil and can be affected by corneal distortion [29].

Optical coherence tomography (OCT) is a non-contact, non-invasive technique that provides high-resolution cross-sectional images of the retina and anterior segment. It is widely considered to be the most accurate method for measuring axial length, but it can be affected by small amounts of corneal reflection and scatter [30]. The Pentacam measures axial length using optical low coherence reflectometry (OLCR) technology. This is a non-contact, non-invasive method of measuring the axial length of the eye. The measurement is performed by shining a light into the eye and analyzing the reflections that are returned. The measurement is typically performed in a matter of seconds, providing a quick and easy method of measuring axial length. The Pentacam has become a commonly used tool in ophthalmology for the assessment of eye anatomy and for guiding surgical procedures such as cataract and refractive surgery [31]. The choice of tool for measuring the axial length of the eye depends on various factors, including the accuracy desired, the presence of any ocular or systemic factors that may affect the measurement, and the need for non-invasive or non-contact methods. It is important to consider the advantages and disadvantages of each tool when choosing the most appropriate method for a specific patient or research study.

The Pentacam offers several advantages over traditional methods for calculating the axial length (AXL) of the eye. Some of these advantages include: non-contact technology [32]: the Pentacam uses a rotating Scheimpflug camera to capture images of the eye, which eliminates the need for contact with the eye and reduces the risk of corneal damage or infection [28]. Second, high precision: the Pentacam uses laser-based technology to accurately measure the AXL, providing highly precise and reliable results [33]. Third, comprehensive analysis: the Pentacam not only measures the AXL but also provides a comprehensive analysis of the anterior and posterior segments of the eye, allowing for a more complete understanding of the ocular structure and function. Finally, automated process: the Pentacam has an automated process that eliminates the need for manual measurements, reducing the risk of measurement errors and providing more accurate results [34].

Our results showed that axial length, mean keratometry, and anterior chamber depth had significant correlations with myopic refractive error, particularly in subjects with axial lengths greater than 25 mm. A longer axial length is typically associated with a greater degree of myopia, but a flatter cornea and deeper anterior chamber depth can reduce the refractive error. A less flat cornea leads to lower refractive power, resulting in the light focusing on a point further from the cornea (i.e., hyperope displacement). When the anterior chamber depth increases, the distance between the cornea and lens increases, which also leads to a hyperopic change. These compensatory changes can result in emmetropization. Our results are consistent with other studies that have found flatter corneas in myopia [24,35].

It is currently unknown if the biological processes underlying the development of myopia at the age of 7 differ from those that occur during the early adult years, so the age of onset of myopia may not be a reliable indicator [2]. It has also been shown that structural complications in individuals with severe myopia are highly dependent on age [36]. Chua et al. found that the age at the onset of myopia, or the duration of its progression, was the most important predictor in cases of severe myopia in myopic children [3]. Our results agree with these findings, since patients who had developed myopia before age 11 had higher values of SE without CP, AXL, and Q. Regarding Q, our study is consistent with that of Horner et al., who suggested that young nearsighted individuals with more prolate corneas tended to develop higher degrees of juvenile myopia [37], as opposed to the study performed by Yebra-Pimentel et al., who found no correlation between Q and refractive error [38]. Although previous studies have reported a significant correlation between PQT and refractive error [39], our results suggest that the correlation between PQT and myopia is very weak. Several population studies have reported that men were slightly but significantly more nearsighted than women [40,41]. In our study, there were no significant differences between men and women for any of the variables.

The strengths of this study include the use of the Pentacam AXL^®^ device, a reliable and accurate tool for measuring ocular biometrics, and the prospective, cross-sectional design, which allowed for the evaluation of the relationship between myopia and ocular biometric variables in a large sample of patients. The study also included a wide range of variables, including maximum and minimum keratometry, anterior chamber depth, corneal horizontal diameter, central corneal thickness, corneal asphericity, and axial length. This design offers several advantages, including time-efficiency, as data are collected at one point in time; low cost, as no follow-up is needed; and suitability for large sample sizes, as the design can accommodate a large sample size. Compared to other methodologies, this design is different as it is prospective, collecting data in advance, and cross-sectional, collecting data at one point in time, instead of retrospectively or repeatedly over time. Additionally, this study was conducted at a single location, unlike multi-center designs which are conducted at multiple locations. The measurements were taken multiple times to minimize error and the quality factor of the tomographic and biometric measurements was required to be greater than 95% to be considered optimal. The results of the study showed significant correlations between axial length and several other variables, including spherical equivalent, age at onset of myopia, mean keratometry, anterior chamber depth, and corneal horizontal diameter. This suggests that axial length is an important factor in the development and progression of myopia and that it may be useful in identifying the most appropriate correction techniques for patients. Overall, the study provides valuable insights into the relationship between myopia and ocular biometric variables, and demonstrates the effectiveness of the Pentacam AXL^®^ in measuring these variables. These findings have important implications for the management of myopia and may inform the development of strategies for its control.

There are several limitations to this study that should be considered when interpreting the results. First, the sample size is relatively small, with only 50 patients included in the study. This may not be representative of the larger population and could affect the generalizability of the findings. Second, the study is cross-sectional in nature, which means that it only captures a snapshot of the data at a single point in time. This precludes the ability to establish any causal relationships between the variables of interest. Third, the study is limited to a single center, which may not be representative of the diversity of patient populations seen in other settings. Finally, the study is limited to a specific age range of 18–30 years, which means that the results may not be applicable to individuals outside of this age range. In addition, all patients reported approximately the age of onset of myopia, but their medical history was not consulted the first time they underwent refraction.

In addition to the limitations already mentioned, there are a few other factors to consider when interpreting the results of this study. First, the sample is restricted to Caucasian patients, which means that the results may not be generalizable to other racial or ethnic groups. Second, the study only included patients with myopia, which means that the results may not be applicable to individuals with other types of refractive errors. Third, the measurement of ocular biometric variables was performed using the Pentacam AXL single rotation Scheimpflug camera, which is a specific type of instrument with its own set of limitations and potential sources of error. Fourth, the cycloplegic refraction was not performed. Cycloplegia is important because it temporarily paralyzes the accommodation reflex of the eye, this is particularly useful in determining the appropriate prescription for corrective lenses [42]. When the spherical equivalent is calculated, several factors should be considered, including the patient’s age, their level of accommodative ability, and any previous prescription for corrective lenses [42]. Finally, the study did not examine the influence of environmental or genetic factors on the relationship between myopia and ocular biometric variables, which could be important considerations in future research.

There are several directions for future research that could build upon the findings of this study. First, it would be interesting to replicate this study in a larger and more diverse sample to confirm the results and determine the generalizability of the findings to other populations. Second, a longitudinal study design, in which the same patients are followed over time, could provide insights into the temporal relationships between myopia and ocular biometric variables. Third, the influence of environmental and genetic factors on the relationship between myopia and ocular biometric variables could be examined in future research. This could be achieved by collecting data on patient exposures and family history and using statistical techniques to control for these factors. Finally, the use of other measurement instruments or techniques, such as ultrasound or magnetic resonance imaging, could provide additional insights into the relationship between myopia and ocular biometric variables.

## 5. Conclusions

AXL is the biometric variable with the greatest influence on the final refractive state in the adult myopic eye. Ophthalmologists and optometric management must consider these biometric differences in order to identify the most appropriate correction techniques in each case. The use of the Pentacam AXL in ocular biometric measurement is effective, reproducible, and non-invasive. The results of our study suggest that the use of the Pentacam device could be an effective tool for monitoring the progression of AXL in myopic patients.

## Figures and Tables

**Figure 1 life-13-00447-f001:**
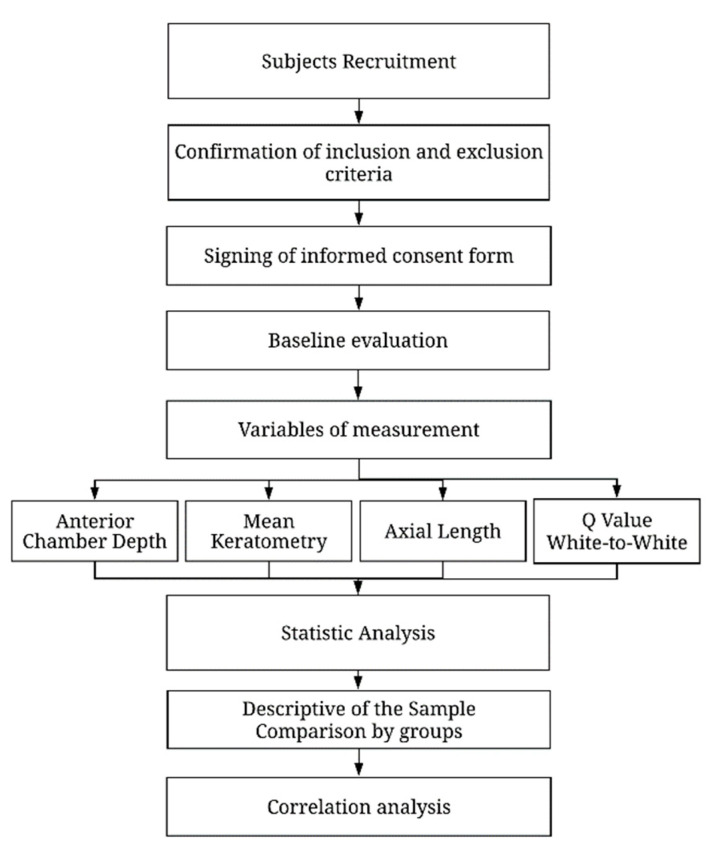
Research procedure flowchart.

**Figure 2 life-13-00447-f002:**
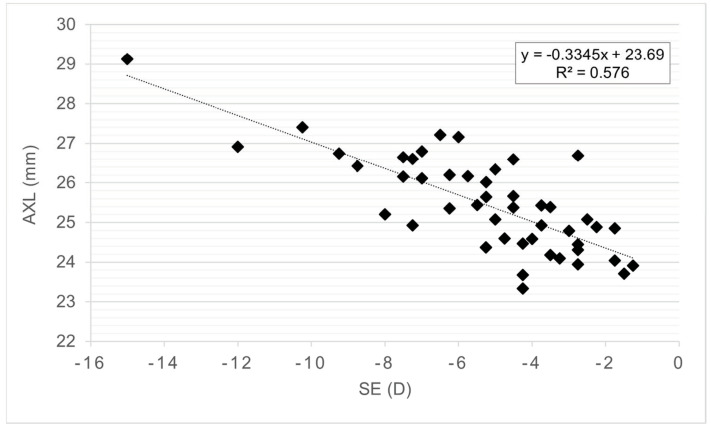
Correlation graph between spherical equivalent and AXL. Axial length (AXL), Spherical equivalent (SE) without cycloplegia and diopters (D).

**Figure 3 life-13-00447-f003:**
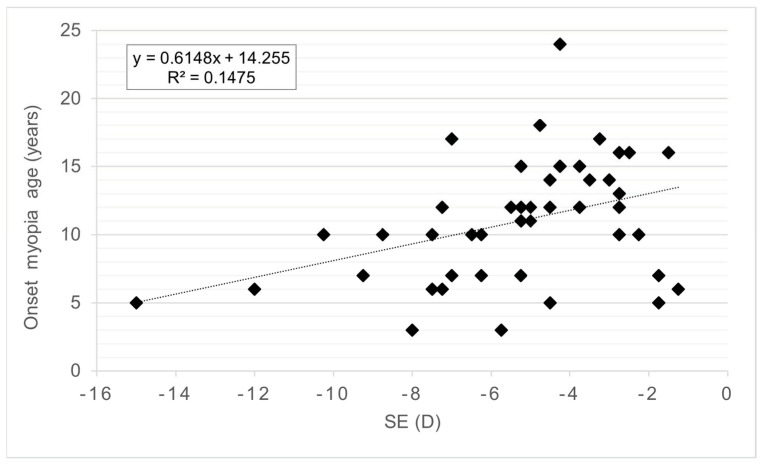
Correlation graph between onset of myopia age and spherical equivalent. Spherical equivalent (SE) without cycloplegia (CP) and diopters (D).

**Table 1 life-13-00447-t001:** Descriptive analysis of the sample.

Variables			
	Mean ± SD	95% CI	
		Lower	Higher
Age (years)	24.84 ± 3.04	24.42	26.12
Onset of Myopia (years)	10.83 ± 4.48	9.68	12.37
SE without CP (D)	−5.16 ± 2.72	−6.05	−4.36
AXL (mm)	25.40 ± 1.20	24.97	25.71
Km (D)	43.87 ± 1.25	43.50	44.28
Q	−0.34 ± 0.10	−0.37	−0.34
Pachymetry (µm)	525.09 ± 30.42	515.95	534.23
ACD (mm)	3.32 ± 0.27	3.23	3.40
WTW (mm)	11.96 ± 0.34	11.85	12.05

SE = Sphere Equivalent; CP = Cycloplegia; AXL = Axial Length; Km = Mean anterior corneal keratometry; Q = Asphericity; ACD = Anterior Chamber Depth; WTW = Horizontal corneal diameter; D = Diopters; mm = millimeters; µm = micrometers; SD = Standard Deviation.

**Table 2 life-13-00447-t002:** Differences in the demographic and ocular parameters between groups by the SE, AXL, and onset of myopia age.

	SE			AXL			Onset Myopia Age			Statistical Test
	≤−4.5D	>−4.5D	*p*	≤25 mm	>25 mm	*p*	≤11 years	>11 years	*p*	
n	28	22		22	28		25	25		
Males/Females (n)	11/17	10/12	0.440	8/14	13/15	0.335	9/16	12/13	0.284	1
Age (years)	24.57 ± 3.27	25.18 ± 2.77	0.488	25.45 ± 2.72	24.36 ± 3.25	0.209	24.92 ± 3.46	24.76 ± 2.63	0.855	2
Onset of Myopia (years)	9.22 ± 3.52	13.00 ± 4.79	0.003 *	9.61 ± 3.31	12.33 ± 5.30	0.037 *	7.44 ± 2.42	14.68 ± 2.83	<0.001 *	2
SE without CP (D)	−6.87 ± 2.43	−2.99 ± 0.90	<0.001 *	−3.44 ± 1.43	−6.51 ± 2.74	<0.001 *	−6.60 ± 2.89	−3.72 ± 1.54	<0.001 *	3
AXL (mm)	26.09 ± 1.04	24.52 ± 0.73	<0.001 *	24.32 ± 0.44	26.25 ± 0.89	<0.001 *	25.96 ± 1.28	24.85 ± 0.82	0.001	2
Km (D)	43.82 ± 1.22	43.80 ± 1.31	0.822	44.43 ± 1.44	43.43 ± 0.87	0.016 *	43.97 ± 1.47	43.77 ± 1.01	0.459	3
Q	−0.35 ± 0.10	−0.32 ± 0.10	0.230	−0.30 ± 0.09	−0.37 ± 0.10	0.023 *	−0.37 ± 0.11	−0.31 ± 0.09	0.047	2
Pachymetry (µm)	526,82 ± 29.67	525.95 ± 29.55	0.919	522.73 ± 29.79	529.36 ± 29.14	0.433	526.36 ± 29.37	526.52 ± 29.86	0.985	2
ACD (mm)	3.35 ± 0.26	3.28 ± 0.28	0.289	3.27 ± 0.25	3.37 ± 0.28	0.214	3.34 ± 0.25	3.31 ± 0.29	0.748	2
WTW (mm)	11.99 ± 0.30	11.93 ± 0.40	0.582	11.83 ± 0.42	12.07 ± 0.22	0.030 *	11.91 ± 0.31	12.01 ± 0.37	0.207	3

SE = Sphere Equivalent; CP = Cycloplegia; AXL = Axial Length; Km = Mean anterior corneal keratometry; Q = Asphericity; ACD = Anterior Chamber Depth; WTW= Horizontal corneal diameter; D = Diopters; mm = millimeters; µm = micrometers. Statistical test performed: Chi-square test (1), Independent sample Student’s *t*-test (2) and Mann–Whitney U test (3). * Statistically significant data (*p* < 0.05).

## Data Availability

The data presented in this study are available on request from the corresponding author. The data are not publicly available due to their containing information that could compromise the privacy of research participants.

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
