# Peer review of "Key Factors in Early Diagnosis of Myopia Progression within Ocular Biometric Parameters by Scheimpflug Technology"

_life, 2023, doi:10.3390/life13020447_

Round 1
Reviewer 1 Report
In this study, authors investigate the relationship between myopia and ocular biometric variables using the Pentacam AXL® single rotation Scheimpflug camera. The authors claimed that the use of Pentacam AXL in visual biometric measurements is effective, reproducible, and non-invasive. However, several flaws should be revised before publication.
What is the main theme of this research in this manuscript? A flowchart or tables that represent the process (sample size, comparisons, analysis) should be included in detail to improve the reader's comprehension.
Authors should supplement more words in their study and the detailed parameters in the process of data analysis.
Please provide more details about the design scheme method and tell readers the difference and advantages of this technique with other approaches.
Authors should compare their results with available tools.
Use a non-parametric test if the assumptions are not met. You can carry out a permutation test (Fisher exact test) that requires no distribution assumptions (require enough computational power). For multiple testing corrections, Benjamini-Hochberg is a widely used FDR algorithm and it has the best FP and FN controlling performance under independence assumptions.
The quality of the figures is not good and needs to be modified.
There are a few minor language issues that need to be addressed, and it is necessary to find a professional retouching company for retouching.
The additional experiments are essential to validate these results.
Brief definitions for "sensitivity" and "specificity" should be added.
Abbreviations in figures should be explained in the figure sub-tag.
References that are not very relevant to the subject of this review should be removed.
Author Response
Reviewer 1
#RV0: In this study, authors investigate the relationship between myopia and ocular biometric variables using the Pentacam AXL® single rotation Scheimpflug camera. The authors claimed that the use of Pentacam AXL in visual biometric measurements is effective, reproducible, and non-invasive. However, several flaws should be revised before publication.
#AU0: We are very grateful for the response received after submitting our Manuscript entitled "Key Factors in Early Diagnosis of Myopia Progression within Ocular Biometric Parameters by Scheimpflug Technology". The authors have considered the comments made by the reviewers and have integrated the necessary corrections in it. Undoubtedly, these contributions will improve the quality of the article. In following, you can find all the answers.
#RV1: What is the main theme of this research in this manuscript? A flowchart or tables that represent the process (sample size, comparisons, analysis) should be included in detail to improve the reader's comprehension.
#AU1: Thank you very much for the comment, we have included a flowchart in the manuscript to improve the reader’s comprehension.
#RV2: Authors should supplement more words in their study and the detailed parameters in the process of data analysis.
#AU2: We agree with your comment, the authors have included additional information in the statistically analysis section.
#RV3: Please provide more details about the design scheme method and tell readers the difference and advantages of this technique with other approaches.
#AU3: Thanks for the comment, we have included the following content in the manuscript:
“This design offers several advantages, including time-efficiency as data is collected at one point in time, low cost as no follow-up is needed, and suitability for large sample sizes as the design can accommodate a large sample size. Compared to other methodologies, this design is different as it is prospective, collecting data prospectively in advance, and cross-sectional, collecting data at one point in time, instead of retrospectively or repeatedly over time. Additionally, this study was conducted at a single location, unlike multi-center designs which are conducted at multiple locations.”
#RV4: Authors should compare their results with available tools.
#AU4: Thanks for the comment. The available tools for axial length measurement were included in the manuscript.
There are various tools available for the measurement of axial length, including ultrasound [1], partial coherence laser interferometry [2], and optical coherence tomography [3]. Each of these tools has its own unique advantages and disadvantages. Ultrasound is a non-invasive method of measuring axial length, but its accuracy can be affected by factors such as the thickness of the cornea, presence of cataracts, and the presence of media opacities [1]. On the other hand, partial coherence laser interferometry provides highly accurate measurements, but requires a dilated pupil and can be affected by corneal distortion [2].
Optical coherence tomography (OCT) is a non-contact, non-invasive technique that provides high-resolution cross-sectional images of the retina and anterior segment. It is widely considered to be the most accurate method for measuring axial length, but it can be affected by small amounts of corneal reflection and scatter [3]. The Pentacam measures axial length using optical low coherence reflectometry (OLCR) technology. This is a non-contact, non-invasive method of measuring the axial length of the eye. The measurement is performed by shining a light into the eye and analyzing the reflections that are returned. The measurement is typically performed in a matter of seconds, providing a quick and easy method of measuring axial length. The Pentacam has become a commonly used tool in ophthalmology for the assessment of eye anatomy and for guiding surgical procedures such as cataract and refractive surgery [4]. The choice of tool for measuring axial length of the eye depends on various factors, including the accuracy desired, the presence of any ocular or systemic factors that may affect the measurement, and the need for non-invasive or non-contact methods. It is important to consider the advantages and disadvantages of each tool when choosing the most appropriate method for a specific patient or research study.
#RV5: Use a non-parametric test if the assumptions are not met. You can carry out a permutation test (Fisher exact test) that requires no distribution assumptions (require enough computational power). For multiple testing corrections, Benjamini-Hochberg is a widely used FDR algorithm and it has the best FP and FN controlling performance under independence assumptions.
#AU5: The authors appreciate your comments. The distribution of the data for each variable was analyzed by the Shapiro-Wilk test. Parametric and non-parametric variables has been defined in the manuscript and the following sentence has been added: “Data distribution normality was studied by the Shapiro-Wilk test. The Chi-square test was used to analyse the contingency table Due to the number of patients and their distribution. However, Fisher's exact test provided the same results. For the multiple regression analysis, a stepwise system was employed to determine the most relevant variables. This has been specified in the manuscript.
#RV6: The quality of the figures is not good and needs to be modified.
#AU6: The authors agree with your comment. The resolution of the images has been improved.
#RV7: There are a few minor language issues that need to be addressed, and it is necessary to find a professional retouching company for retouching.
#AU7: We appreciate your comments. An English revision of the text has been performed. We send certificate of language editing services.
#RV8: The additional experiments are essential to validate these results.
#AU8: Thanks for the comment, we have included this limitation in the section of limitations in the manuscript.
#RV9: Brief definitions for "sensitivity" and "specificity" should be added.
#AU9: The authors agree with your comment. Definitions have been included.
“When talking about a clinical diagnostic test, parameters such as sensitivity, specificity, and positive and negative predictive values are described. These reflect the characteristics of a diagnostic test and are used to decide when they should be used (sensitivity and specificity of a test) or what meaning a test result has in a particular patient.
Sensitivity is the probability of correctly classifying patients or, what is the same, the proportion of true positives. While the specificity is the probability of correctly classifying the healthy ones or, what is the same, the proportion of true negatives. Accordingly, the sensitivity and specificity represent the validity of a diagnostic test, and the positive predictive value and negative predictive value represent the safety of a diagnostic test [5,6].”
#RV10: Abbreviations in figures should be explained in the figure sub-tag.
#AU10: Thanks for the comment, we have included the abbreviation explanation.
#RV11: References that are not very relevant to the subject of this review should be removed.
#AU11: We agree with your comment, the references that do not link with the subjects were removed.
- Trivedi, R.H.; Wilson, M.E. Globe axial length data in children using immersion A-scan ultrasound. J. Cataract Refract. Surg. 2021, 47, 1481–1482, doi:10.1097/j.jcrs.0000000000000527.
- Roessler, G.F.; Talab, Y.D.; Dietlein, T.S.; Dinslage, S.; Plange, N.; Walter, P.; Mazinani, B.A. Partial Coherence Laser Interferometry in Highly Myopic versus Emmetropic Eyes. J. Ophthalmic Vis. Res. 2014, 9, 169–173.
- Chirapapaisan, C.; Srivannaboon, S.; Chonpimai, P. Efficacy of Swept-source Optical Coherence Tomography in Axial Length Measurement for Advanced Cataract Patients. Optom. Vis. Sci. Off. Publ. Am. Acad. Optom. 2020, 97, 186–191, doi:10.1097/OPX.0000000000001491.
- Cooke, D.L.; Cooke, T.L. Approximating sum-of-segments axial length from a traditional optical low-coherence reflectometry measurement. J. Cataract Refract. Surg. 2019, 45, 351–354, doi:10.1016/j.jcrs.2018.12.026.
- Monaghan, T.F.; Rahman, S.N.; Agudelo, C.W.; Wein, A.J.; Lazar, J.M.; Everaert, K.; Dmochowski, R.R. Foundational Statistical Principles in Medical Research: Sensitivity, Specificity, Positive Predictive Value, and Negative Predictive Value. Medicina (Kaunas). 2021, 57, doi:10.3390/MEDICINA57050503.
- Binney, N.; Hyde, C.; Bossuyt, P.M. On the Origin of Sensitivity and Specificity. Ann. Intern. Med. 2021, 174, 401–407, doi:10.7326/M20-5028.
Reviewer 2 Report
This manuscript is an interesting work on myopia, the statistical analysis is correct.
Unfortunately, the refractive error was measured without cycloplegia. However, accommodation is well known to be a very important factor in the development of actual refractive power, especially in young adults. Were these subjects truly myopic?
The authors are recommended to update the references (the latest one was published in 2019) and to review the discussion.
(Clinical Validation of a New Optical Biometer for Myopia Control in a Healthy Pediatric Population by Elena Martínez-Plaza,Ainhoa Molina-Martín,Alfonso Arias-Puente and David P. Piñero Children 2022, 9(11), 1713; https://doi.org/10.3390/children9111713 - 09 Nov 2022
Mathematical Estimation of Axial Length Increment in the Control of Myopia Progression by António Queirós,Ana Amorim-de-Sousa,Paulo Fernandes,Maria Sameiro Ribeiro-Queirós,César Villa-Collar andJosé M. González-Méijome J. Clin. Med. 2022, 11(20), 6200; https://doi.org/10.3390/jcm11206200 - 20 Oct 2022
etc)
The impact of AXL on the progression of myopia is well known. Focus on the different methods and discuss their values.
Author Response
Reviewer 2
#RV0: This manuscript is an interesting work on myopia, the statistical analysis is correct.
#AU0: We are very grateful for the response received after submitting our Manuscript entitled "Key Factors in Early Diagnosis of Myopia Progression within Ocular Biometric Parameters by Scheimpflug Technology". The authors have considered the comments made by the reviewers and have integrated the necessary corrections in it. Undoubtedly, these contributions will improve the quality of the article. In following, you can find all the answers.
#RV1: Unfortunately, the refractive error was measured without cycloplegia. However, accommodation is well known to be a very important factor in the development of actual refractive power, especially in young adults. Were these subjects truly myopic?
#AU1: Thank you very much for the comment, the presence of myopia was confirmed by performing retinoscopy and fogging [1,2]. However we have included this issue on the limitation.
#RV2: The authors are recommended to update the references (the latest one was published in 2019) and to review the discussion.
(Clinical Validation of a New Optical Biometer for Myopia Control in a Healthy Pediatric Population by Elena Martínez-Plaza,Ainhoa Molina-Martín,Alfonso Arias-Puente and David P. Piñero Children 2022, 9(11), 1713; https://doi.org/10.3390/children9111713 - 09 Nov 2022
Mathematical Estimation of Axial Length Increment in the Control of Myopia Progression by António Queirós,Ana Amorim-de-Sousa,Paulo Fernandes,Maria Sameiro Ribeiro-Queirós,César Villa-Collar andJosé M. González-Méijome J. Clin. Med. 2022, 11(20), 6200; https://doi.org/10.3390/jcm11206200 - 20 Oct 2022 etc)
#AU2: We appreciate your comments. References have been included [3,4].
#RV3: The impact of AXL on the progression of myopia is well known. Focus on the different methods and discuss their values.
#AU3: We agree with your comment, we have included a comparison between the different methods of measurements.
“There are various tools available for the measurement of axial length, including ultrasound [5], partial coherence laser interferometry [6], and optical coherence tomography [7]. Each of these tools has its own unique advantages and disadvantages. Ultrasound is a non-invasive method of measuring axial length, but its accuracy can be affected by factors such as the thickness of the cornea, presence of cataracts, and the presence of media opacities [5]. On the other hand, partial coherence laser interferometry provides highly accurate measurements, but requires a dilated pupil and can be affected by corneal distortion [6].
Optical coherence tomography (OCT) is a non-contact, non-invasive technique that provides high-resolution cross-sectional images of the retina and anterior segment. It is widely considered to be the most accurate method for measuring axial length, but it can be affected by small amounts of corneal reflection and scatter [7]. The Pentacam measures axial length using optical low coherence reflectometry (OLCR) technology. This is a non-contact, non-invasive method of measuring the axial length of the eye. The measurement is performed by shining a light into the eye and analyzing the reflections that are returned. The measurement is typically performed in a matter of seconds, providing a quick and easy method of measuring axial length. The Pentacam has become a commonly used tool in ophthalmology for the assessment of eye anatomy and for guiding surgical procedures such as cataract and refractive surgery [8]. The choice of tool for measuring axial length of the eye depends on various factors, including the accuracy desired, the presence of any ocular or systemic factors that may affect the measurement, and the need for non-invasive or non-contact methods. It is important to consider the advantages and disadvantages of each tool when choosing the most appropriate method for a specific patient or research study.”
- Mukash, S.N.; Kayembe, D.L.; Mwanza, J.C. Agreement Between Retinoscopy, Autorefractometry and Subjective Refraction for Determining Refractive Errors in Congolese Children. Clin. Optom. 2021, 13, 129–136, doi:10.2147/OPTO.S303286.
- Esteves Leandro, J.; Meira, J.; Ferreira, C.S.; Santos-Silva, R.; Freitas-Costa, P.; Magalhães, A.; Breda, J.; Falcão-Reis, F. Adequacy of the Fogging Test in the Detection of Clinically Significant Hyperopia in School-Aged Children. J. Ophthalmol. 2019, 2019, doi:10.1155/2019/3267151.
- Martínez-Plaza, E.; Molina-Martín, A.; Arias-Puente, A.; Piñero, D.P. Clinical Validation of a New Optical Biometer for Myopia Control in a Healthy Pediatric Population. Child. (Basel, Switzerland) 2022, 9, 1713, doi:10.3390/CHILDREN9111713.
- Queirós, A.; Amorim-de-Sousa, A.; Fernandes, P.; Ribeiro-Queirós, M.S.; Villa-Collar, C.; González-Méijome, J.M. Mathematical Estimation of Axial Length Increment in the Control of Myopia Progression. J. Clin. Med. 2022, 11, doi:10.3390/JCM11206200.
- Trivedi, R.H.; Wilson, M.E. Globe axial length data in children using immersion A-scan ultrasound. J. Cataract Refract. Surg. 2021, 47, 1481–1482, doi:10.1097/j.jcrs.0000000000000527.
- Roessler, G.F.; Talab, Y.D.; Dietlein, T.S.; Dinslage, S.; Plange, N.; Walter, P.; Mazinani, B.A. Partial Coherence Laser Interferometry in Highly Myopic versus Emmetropic Eyes. J. Ophthalmic Vis. Res. 2014, 9, 169–173.
- Chirapapaisan, C.; Srivannaboon, S.; Chonpimai, P. Efficacy of Swept-source Optical Coherence Tomography in Axial Length Measurement for Advanced Cataract Patients. Optom. Vis. Sci. Off. Publ. Am. Acad. Optom. 2020, 97, 186–191, doi:10.1097/OPX.0000000000001491.
- Cooke, D.L.; Cooke, T.L. Approximating sum-of-segments axial length from a traditional optical low-coherence reflectometry measurement. J. Cataract Refract. Surg. 2019, 45, 351–354, doi:10.1016/j.jcrs.2018.12.026.
Reviewer 3 Report
For the authors
The manuscript showed the authors important and strong messages, these are great knowledges for the ophthalmologist.
There was one thing to address.
Onset Myopia age is most important factor in the manuscript. But the author doesn’t describe how they correct this information in the method.
I thought it is difficult to decide when it happened and felt it’s inaccurate data the authors decided Onset Myopia age. If the authors asked from patients, probably many patients remembered like about 6 to 9 years their myopia happened. But the authors data showed when it happened. It will be needed precisely how the authors corrected and defined Onset Myopia age of patients. It’s very important information to describe in the manuscripit.
Author Response
Reviewer 3
#RV0: The manuscript showed the authors important and strong messages, these are great knowledges for the ophthalmologist.
#AU0: We are very grateful for the response received after submitting our Manuscript entitled "Key Factors in Early Diagnosis of Myopia Progression within Ocular Biometric Parameters by Scheimpflug Technology". The authors have considered the comments made by the reviewers and have integrated the necessary corrections in it. Undoubtedly, these contributions will improve the quality of the article. In following, you can find all the answers.
#RV1: There was one thing to address. Onset Myopia age is most important factor in the manuscript. But the author doesn’t describe how they correct this information in the method. I thought it is difficult to decide when it happened and felt it’s inaccurate data the authors decided Onset Myopia age. If the authors asked from patients, probably many patients remembered like about 6 to 9 years their myopia happened. But the authors data showed when it happened. It will be needed precisely how the authors corrected and defined Onset Myopia age of patients. It’s very important information to describe in the manuscript.
#AU1: The authors agree with your comment. All patients reported approximately the age of onset of myopia, but their medical history was not consulted the first time they underwent refraction. It has been included as a limitation in the research.
Round 2
Reviewer 2 Report
Please find attachment

Author Response
Reviewer 2
#AU0: We are very grateful for the response received after submitting our Manuscript entitled "Key Factors in Early Diagnosis of Myopia Progression within Ocular Biometric Parameters by Scheimpflug Technology". The authors have considered the comments made by the reviewers and have integrated the necessary corrections in it. Undoubtedly, these contributions will improve the quality of the article.
In following, you can find all the answers.
#RV1: The main problem is that spherical equivalent was calculated without cycloplegia. The authors should prove their patients were truly myopic: it should be described in the section Methods.
#AU1: Thanks for the comment, the participants underwent without cycloplegic autorefraction and traditional refraction using retinoscopy and then subjective refraction with fogging was performed.
“Fogging refers to using plus powers to bring the optical point of focus in front of the retina to ensure that accommodation is adequately relaxed. The principle of fogging involves using spherical powers to create artificial myopia, thereby moving the entire area of focus in the eye in front of the retina to create a situation where an attempt at accommodating will blur the vision, which further causes the patient to relax accommodation. Fogging is effective irrespective of the inherent refractive state of the eye and the efficacy of fogging in refraction has been demonstrated [1].”
#RV2: All of the statistical descriptions in the text, tables, figures should indicate spherical equivalent measured without cycloplegia. (SE without CP)
#AU2: Thanks for the comment, we have included SE without CP in the text, tables and figures.
#RV3: Discussion: Only one sentence could be found in the limitations (L349): Fourth, the cycloplegic refraction was not performed. It is not enough. Please describe why cycloplegia is important, which factors should be considered when spherical equivalent is calculated. Without cycloplegia spherical equivalent does not show the true degree of myopia. Any correlation with SE measured without CP is only a rough estimation. These should be indicated in the manuscript, as well.
#AU3: Thanks for the comment, we totally agree with your appreciation, we have included in the manuscript:
“Cycloplegia is important because it temporarily paralyzes the accommodation reflex of the eye, this is particularly useful in determining the appropriate prescription for corrective lenses [2]. When spherical equivalent is calculated, several factors should be considered, including the patient's age, their level of accommodative ability, and any previous prescription for corrective lenses [2].”
#RV4: It is not new, that SE is correlated with AXL in axial type myopia. The value of Pentacam over other methods is highly recommended to discuss. The most important message is that Pentacam could be a useful tool in the screening of AXL progression among myopic patients.
#AU4: We agree with your comment, we have included in the discussion section (included in Line 280):
“The Pentacam offers several advantages over traditional methods for calculating the axial length (AXL) of the eye. Some of these advantages include Non-contact technology [3]: The Pentacam uses a rotating Scheimpflug camera to capture images of the eye, which eliminates the need for contact with the eye and reduces the risk of corneal damage or infection [4]. Second, high precision: The Pentacam uses laser-based technology to accurately measure the AXL, providing highly precise and reliable results [5]. Third, comprehensive analysis: The Pentacam not only measures the AXL but also provides a comprehensive analysis of the anterior and posterior segments of the eye, allowing for a more complete understanding of the ocular structure and function. Finally, Automated process: The Pentacam has an automated process that eliminates the need for manual measurements, reducing the risk of measurement errors and providing more accurate results [6].”
In addition, we have included that the important message that Pentacam could be useful tool in the screening of AXL progression among myopic patients.
We have included in the conclusion:
“The results of our study suggest that the use of the Pentacam device could be an effective tool for monitoring the progression of AXL in myopic patients.”
- Manna, P.; Karmakar, S.; Bhardwaj, G.K.; Mondal, A. Accommodative spasm and its different treatment approaches: A systematic review. Eur. J. Ophthalmol. 2022, 112067212211364, doi:10.1177/11206721221136438.
- Ampolu, N.; Yarravarapu, D.; Satgunam, P.; Varadharajan, L.S.; Bharadwaj, S.R. Impact of induced pseudomyopia and refractive fluctuations of accommodative spasm on visual acuity. Clin. Exp. Optom. 2022, 1–7, doi:10.1080/08164622.2022.2140583.
- Yu, J.; Wen, D.; Zhao, J.; Wang, Y.; Feng, K.; Wan, T.; Savini, G.; McAlinden, C.; Lin, X.; Niu, L.; et al. Comprehensive comparisons of ocular biometry: A network-based big data analysis. Eye Vis. (London, England) 2022, 10, 1, doi:10.1186/s40662-022-00320-3.
- Trivedi, R.H.; Wilson, M.E. Globe axial length data in children using immersion A-scan ultrasound. J. Cataract Refract. Surg. 2021, 47, 1481–1482, doi:10.1097/j.jcrs.0000000000000527.
- Kanclerz, P.; Hoffer, K.J.; Bazylczyk, N.; Wang, X.; Savini, G. Optical Biometry and IOL Calculation in a Commercially Available Optical Coherence Tomography Device and Comparison With Pentacam AXL. Am. J. Ophthalmol. 2023, 246, 236–241, doi:10.1016/j.ajo.2022.09.022.
- Moshirfar, M.; Tenney, S.; McCabe, S.; Schmid, G. Repeatability and reproducibility of the galilei G6 and its agreement with the pentacam® AXL in optical biometry and corneal tomography. Expert Rev. Med. Devices 2022, 19, 375–383, doi:10.1080/17434440.2022.2075725.
Reviewer 3 Report
The authors added their sentences in limitation. The manuscript has the potential to be published.
Author Response
Thank you very much for your time and dedication in the review of this manuscript.
You have improve all your considerations